# Repairing Damaged Screen Pipes with Tube Hydroforming: Experiments and Feasibility Analysis

**Shufeng Liu [1,2], Hanxiang Wang [1], Wenjian Lan [1,*], Yanxin Liu [1], Jiaqi Che [1] and Shaohua Ma [1]**

[1] College of Mechanical and Electronic Engineering, China University of Petroleum (East China), Qingdao 266580, China; b18040125@s.upc.edu.cn (S.L.); wanghx@upc.edu.cn (H.W.); liuyanxin1985@upc.edu.cn (Y.L.); b15040127@s.upc.edu.cn (J.C.); mashaohua85@upc.edu.cn (S.M.)

[2] College of Mechanical and Electronic Engineering, Shandong Agricultural University, Tai'an 271018, China

[*] Correspondence: b15040128@s.upc.edu.cn

**Abstract:** During oil-well production, there are often cracks, breaks, and perforation corrosion on the screen pipe that can significantly deteriorate sand control and pipe strength. To repair damaged screen pipes, we developed a technique originating from the tube hydroforming, and the feasibility of the technique was systematically investigated. First, the elastoplastic mechanics of patch tubes during the hydroforming process was analyzed to investigate the forming mechanism. Second, tensile experiments showed that AISI 321 after cold drawn and solution had good mechanical properties. A numerical simulation model of a hydroforming patch composed of AISI 321 steel was built to investigate the effect of structural parameters such as the length, initial outer diameter, and thickness of a patch tube on hydroforming patch performance. Forming pressure did not significantly change with length, but it decreased with initial outer diameter and increased with thickness. In addition to the simulation, a hydroforming test bench was constructed to experimentally test the patch method. Test results showed that the patch tube could fit closely with the screen base pipe, and residual contact stress could be more than 139.78 kN/m$^2$. Deformation strengthening due to the deformed martensite was conducive to improving the strength of the patch tube after hydroforming. The combination of the simulation and experiment indicates that the application of hydroforming patch technology can effectively repair damaged screen pipes.

**Keywords:** screen pipe; tube hydroforming; patch tube; material property; residual stress

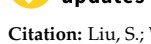



## 1. Introduction

Installing a screen pipe is one of the main methods for developing a horizontal oil well [1–3]. With the long-term development of oil fields, especially with the use of acid pickling and multiple steam injection for mining in a heavy oil reservoir, the well can be affected by long-well section drainage, uneven stream absorption, and local high-speed erosion. Thus, the wall of a sand-control screen pipe can easily crack or break [4–6]. It can become perforated by corrosion or experience some other type of damage, all of which cause the screen pipe to become out of order, leading to the failure of sand control, which causes difficulties in plugging water, the production of a large amount of sand, and serious economic loss [7]. The treatment for sand-control failure mainly focuses on prevention, so developing methods for repairing a damaged screen pipe is urgent [8–10]. However, there is a lack of mature techniques and matching materials for the patch tube used for internally repairing a screen pipe damaged by breaks or corrosion perforation [11]. There are mainly two temporary repair methods: hanging a small-diameter sand screen on the pipe for sand control or installing a packer. Both methods have a short validity period and small internal diameter [12].

Expandable tube technology is widely used to repair damaged casing, mainly including repairing casing damaged wells with high-expansion-ratio materials such as solid

expandable tubular (SET) or expandable profile liner (EPL) technology, and the sand control construction of horizontal wells with an expandable sand screen (ESS) [13–15]. Al-Abri et al. [16] developed a mathematical model to describe the dynamics of the stick–slip phenomenon during the expansion of a solid tube. Xu et al. [17] experimentally investigated the effect of expansion on various aspects of an expandable J55 steel tube, such as deformation, residual stress, mechanical properties, and the microstructure. Park et al. [18] presented two key ways to improve the expandability or circumferential ductility of a pipe produced with high-Mn steel to improve its applicability. Zhao et al. [19] simulated and studied the feasibility of an expandable profile liner for plugging leaks in directional sections of deep and ultradeep wells. Zhou et al. [20] established analytical and finite element models to investigate the forming behavior of a solid expandable tube based on the twin shear-stress yield criterion. Chen et al. [21] established a mechanical equilibrium equation for the expansion process and a mechanical model of an (empty set) 244.5 mm × (empty set) 177.8 mm expandable liner hanger using a finite element simulation. They analyzed the hanging mechanism and the changes in the mechanical parameters during expansion. Shi et al. [22] studied the failure mechanisms of a solid expandable tubular by establishing a finite element model of a defective expandable tubular in a laboratory experiment. Zhao and Duan [23] obtained the internal pressure strength and collapse strength of an expandable profile liner by simulating and analyzing the expansion of three different materials and three different wall thicknesses. In summary, most techniques based on an expandable tube mainly aim to repair a damaged casing, and there are still few methods for repairing damaged screen pipes with an expandable tube.

Hydroforming is widely used in automotive, aerospace, and other fields to realize the plastic deformation of parts through hydraulic pressure [24,25]. In this paper, on the basis of the requirements of a downhole screen pipe, we designed a hydroforming patch technology to repair a damaged screen pipe. As shown in Figure 1, the steps in hydroforming patch technology are as follows:

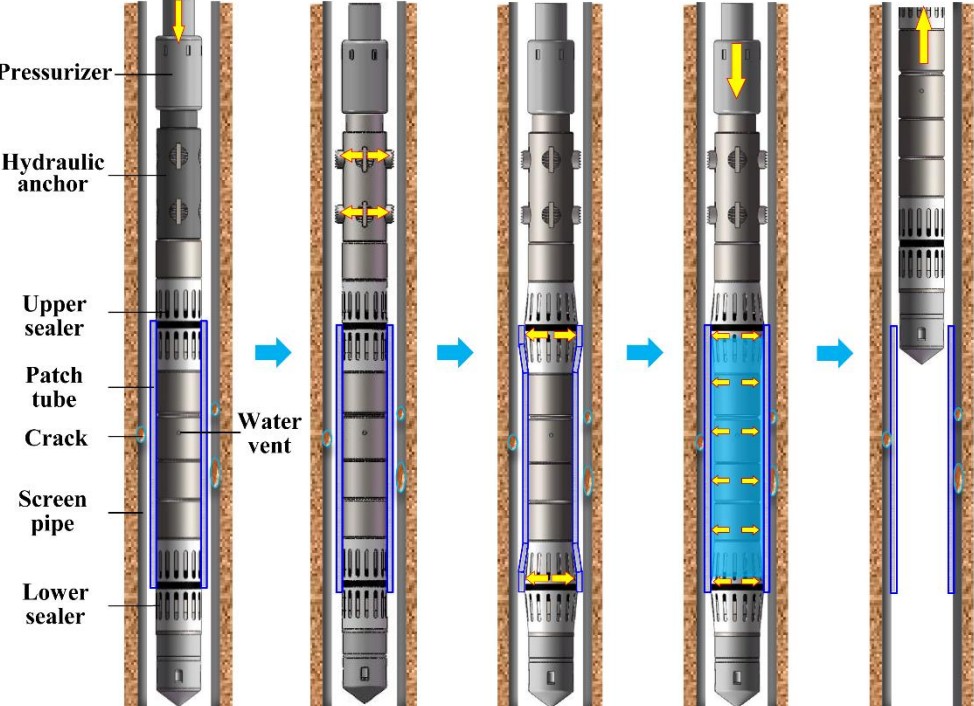

**Figure 1.** Structure of the patch tool.

(1) The patch tool and patch tube are lowered into the damaged screen tube. The structure of the patch tool is shown in Figure 1. (2) The fracturing pump truck supplies hydraulic power, and the hydraulic anchor fixes the patch tool onto the inner wall of the screen pipe.

(3) The downhole pressurizer increases the water pressure, and the down sealer expands and deforms the lower end of the patch tube to the shape of the inner wall of the screen pipe. (4) The water vent in the patch tool is opened. The space between the patch tube and patch tool fills with water, and the air is discharged. (5) The upper sealer expands and deforms the upper end of the patch tube to the shape of the inner wall of the screen tube. The water vent continues to inject high-pressure water, and the patch tube expands and attaches to the inner wall of the screen tube. (6) Pressure is maintained for a set time. After decompression, the patch tool is lifted out.

Hydroforming patch technology can produce reliable bonding between the screen pipe and the patch tube. It has the technical advantages of a large internal diameter, long validity period, and low cost. AISI 321 was selected in a previous study as the material for the patch tube. In the present paper [26], theoretical and practical research is carried out to assess the formability of the material and the patching process. Results may provide a reference for patch hydroforming in repairing damaged screen pipes.

## 2. Analytical Study

The description of a two-layer tube can provide reference for the analysis of hydroforming patch technology. Alexandrov et al. [27] provided a simple analytical solution for describing the expansion of a two-layer tube under plane-strain conditions. Cheng et al. [28] proposed a finite element method combining a spatial beam element and a two-layer contact gap element, and introduced a virtual transient dynamic method to solve the static buckling of concentric tubular columns. Hu et al. [29] established the expression of the yield pressure of a two-layer pipe, and clarified the mechanism of wrinkling and stabilizing an inner tube. In order to simplify and vividly describe the hydroforming process of a patch tube, von Mises and Tresca yield criteria were selected to deduce yield pressure in this paper.

### 2.1. Fundamental Assumptions

During patch hydroforming, the patch tube is subjected to internal pressure and the contact pressure of the screen base pipe. Both ends of the patch tube are also fixed by patch tools. The calculation is simplified as follows, with the expectation that it still meets accuracy requirements. It was assumed that the patch tube and screen base pipe are an ideal elastic–plastic model, and that the patch tube had a thin wall. If the patch tube is long enough, it can be assumed that the state of stress is plane stress, such that $\sigma_1 = 0$ [30]. $\Delta$ is the gap size between the patch tube and screen base pipe. When $\Delta$ is larger than a critical value, the patch tube would enter the plastic zone before it is in contact with the screen base pipe. It is assumed that the patch tube is thin enough that it enters the plastic zone entirely once the patch tube yields. During patching, the fixed force at both ends of the patch tube is small so the effects of the axial force can be ignored. The volume of the patch tube is considered to be incompressible in all stages of forming [31,32]. This analytical method adopts the Tresca yield criterion.

The initial state of the screen base pipe and patch tube is shown in Figure 2. The initial clearance is $\Delta$. A hydroforming patch is shown in Figure 3. O–0–1–2–3–4–5 is the curve between the internal pressure $P_i$ and the hoop strain of the outer wall of the patch tube $\varepsilon_{\theta ro}$; O'–3'–4'–5' is the curve between the contact pressure $P_c$ and the hoop strain of the inner wall of the screen base pipe $\varepsilon_{\theta Ri}$. The patch tube elastically deforms in section O–0. When pressure increased to point 0, the inner wall of the patch tube first began to yield and entered the elastic–plastic stage. When pressure increased to point 1, the outer wall of the patch tube yielded and entered the full-yield stage. When it reached point 2, the outer wall of the patch tube just touched the inner wall of the screen base pipe. The internal hydraulic force was then transmitted to the screen base pipe through contact pressure $P_c$. Elastic deformation occurred in the screen base pipe, and the two tubes always fit together with the same strain position [33,34].

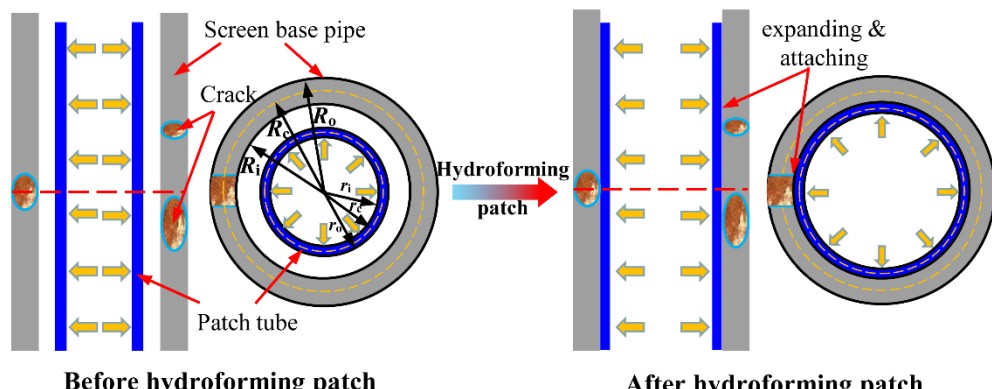

**Figure 2.** Hydroforming patch technology.

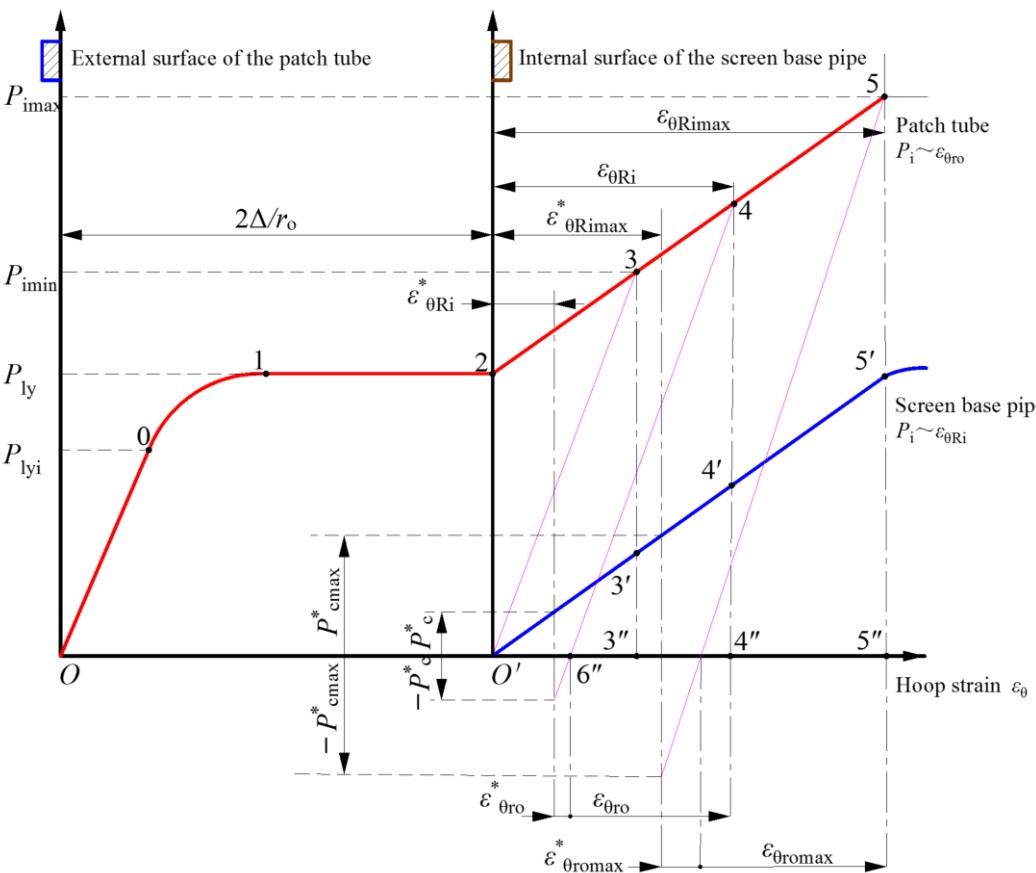

**Figure 3.** Hydroforming patch of patch tube and screen pipe.

### 2.2. Elastoplastic Mechanical Analysis of Hydroforming Patch

When the patch tube was in region O–0, it only underwent elastic deformation. Stress and deformation can be calculated with the Lamé equation [33]:

$$
\begin{cases}
\sigma_{\mathrm{tr}} = \dfrac{P_i}{k_0^2 - 1}\left(1 - \dfrac{r_o^2}{r^2}\right) \\[2mm]
\sigma_{\theta r} = \dfrac{P_i}{k_0^2 - 1}\left(1 + \dfrac{r_o^2}{r^2}\right) \\[2mm]
u = \dfrac{P_i r}{E_1\left(k_0^2 - 1\right)}\left[(1 - \mu_1) + (1 + \mu_1)\dfrac{r_o^2}{r^2}\right]
\end{cases}
\tag{1}
$$

where, $\sigma_{\mathrm{tr}}$ is radial stress in the deformation stage of the patch tube, MPa. $\sigma_{\theta r}$ is hoop stress in the deformation stage of the patch tube, MPa. $P_i$ is the internal pressure of the patch tube, MPa. $u$ is the radial displacement of the patch tube, mm. $k_0$ is the outer–inner diameter

ratio of the patch tube, $k_0 = \frac{r_o}{r_i}$. $\mu_1$ is the Poisson's ratio of the patch tube, and $E_1$ is the elastic modulus of the patch tube, MPa.

When pressure reached point 0, the inner wall of the patch tube began to yield. According to the Tresca yield law, $\sigma_\theta - \sigma_t = \sigma_s$. Updike et al. [35] first proposed the concept of equivalent yield strength. Considering the strain strengthening of the material, material equivalent yield strength $\sigma_{seql}$ was used instead of yield strength $\sigma_s$, where $\sigma_{seql} = \sigma_{tr|r=r_i} > \sigma_s$. Using equivalent yield strength to properly modify the yield strength of materials can improve calculation accuracy. Equivalent yield strength is obtained according to the true stress–strain curve for the patch tube material. The initial yield pressure and radial displacement of the patch tube outer wall are

$$P_{lyi} = \frac{\sigma_{seql}(k_0^2 - 1)}{2k_0^2}, \ u_{ro1} = \frac{r_o \sigma_{seql}}{E_1 k_0^2} \tag{2}$$

where, $\sigma_{seql}$ is the equivalent yield strength of the patch tube, MPa.

From point 0 to point 1, with the increase in internal pressure, the elastic zone decreases and the plastic zone increases. The elastoplastic interface is $r = r_c$, region $r_c < r \leq r_o$ is in the elastic regime, $r_i < r \leq r_c$ is in the plastic regime, and pressure on the interface is $P$, as shown in Figure 4.

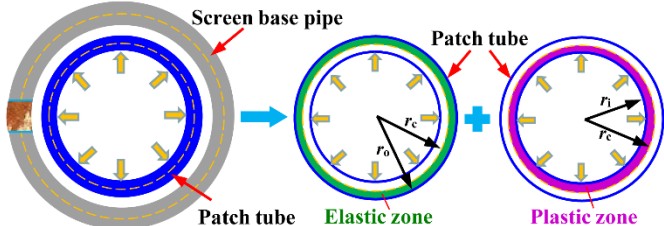

**Figure 4.** Elastic–plastic mechanical analysis of patch tube.

Stress and deformation in the elastic zone of the patch tube can be calculated with the Lamé equation:

$$\begin{cases} \sigma_{tr} = \frac{P' r_c^2}{r_o^2 - r_c^2}\left(1 - \frac{r_o^2}{r^2}\right) \\ \sigma_{\theta r} = \frac{P' r_c^2}{r_o^2 - r_c^2}\left(1 + \frac{r_o^2}{r^2}\right) \\ u = \frac{P' r_c^2 r}{E_1(r_o^2 - r_c^2)}\left[(1 - \mu_1) + (1 + \mu_1)\frac{r_o^2}{r^2}\right] \end{cases} \tag{3}$$

In region $r = r_c$, interface pressure $P'$ is

$$P' = \frac{\sigma_{seql}(r_o^2 - r_c^2)}{2r_o^2} \tag{4}$$

The plastic zone of the patch tube is affected by internal forming pressure $P_i$ and interface pressure $P'$. Due to the balance equation of force $d\sigma_{tr}/dr + (\sigma_{tr} - \sigma_{\theta r}) = 0$, $\sigma_{tr|r=r_i} = -P_i$, $\sigma_{tr|r=r_c} = -P'$. According to the Tresca yield criterion, the mechanical analysis of the plastic region is as follows:

$$\begin{cases} \sigma_{tr} = -P_i - \sigma_{seql} \ln \frac{r}{r_i} \\ \sigma_{\theta r} = -P_i + \sigma_{seql}\left(\ln \frac{r}{r_i} + 1\right) \end{cases} \tag{5}$$

When it reached point 1, the patch tube entered the full-yield state, that is, the patch tube outer wall yielded. The full yield pressure $P_{ly}$ and radial displacement of the patch tube outer wall are

$$P_{ly} = \sigma_{seql} \ln k_0, \ u_{ro2} = \frac{r_o \sigma_{seql}}{E_1} \tag{6}$$

Due to $\Delta$ being larger than $u_{ro2}$, from points 1 to 2, the patch tube fully yielded, and the clearance between the two tubes was closed until the patch tube had touched the screen base pipe. According to Equation (5), from the boundary condition $\sigma_{tr|r=r_o} = -P_c$:

$$P_i = P_{ly} + P_c \tag{7}$$

The screen base pipe would still be in the elastic zone as internal pressure reached $P_i$, and by Hooke's law

$$\sigma_{lR} = \mu_2(\sigma_{tR} + \sigma_{\theta R}), \tag{8}$$

the screen base pipe should not plastically deform during repair operation. The nonassociated flow rule was employed here for simplicity in which the von Mises and Tresca yield criteria were employed to describe the yield potential (for the normality rule) and the yield function (for the elasticity boundary) [36]. Von Mises criterion leads to Equation (9):

$$\sigma_{lr} = \frac{1}{2}(\sigma_{tr} + \sigma_{\theta r}) \tag{9}$$

Therefore, Equations (8) and (9) show that, for both the patch tube and screen base pipe, axial (intermediate principal) stress had no influence on the application of the Tresca criterion.

For O´–6´, if the screen base pipe elastically deformed, then the contact pressure at which the screen base pipe is just about to deform plastically is taken as the maximal contact pressure $P_{cmax}$ between the two tubes:

$$P_{cmax} = \frac{\sigma_{sb}\left(K_0{}^2 - 1\right)}{2K_0{}^2} \tag{10}$$

where, $\sigma_{sb}$ is the yield strength of the screen base pipe, MPa, and $K_0$ is the outer–inner diameter ratio of the screen base pipe, $K_0 = \frac{R_o}{R_i}$. According to the requirements for patching, the plastic deformation of the screen base pipe does not occur after patching is complete. That is, internal forming pressure $P_i$ needs to meet boundary condition $P_i < P_{ly} + P_{cmax}$. Thus, according to Equations (6), (7) and (10):

$$P_i < \sigma_{seql} \ln k_0 + \frac{\sigma_{sb}\left(K_0{}^2 - 1\right)}{2K_0{}^2} \tag{11}$$

*2.3. Residual Contact Stress after Patching*

The outer wall of the patch tube is simultaneously subjected to internal pressure $P_i$ and contact pressure $P_c$ during the forming process:

$$\begin{cases} \sigma_{tro} = -P_c \\ \sigma_{\theta ro} = \sigma_{seql} - P_c \\ P_i - P_c = \sigma_{seql} \ln k_0 \end{cases} \tag{12}$$

where, $\sigma_{tro}$ is radial stress for the outer wall of the patch tube, MPa. $\sigma_{\theta ro}$ is the hoop stress for the outer wall of the patch tube, MPa.

According to the generalized Hooke's law, the stress–strain relationship at the monomer on the outer wall of the patch tube can be described as

$$\begin{cases} \varepsilon_{lro} = \frac{1}{E_1}[\sigma_{lro} - \mu_1(\sigma_{\theta ro} + \sigma_{tro})] \\ \varepsilon_{\theta ro} = \frac{1}{E_1}[\sigma_{\theta ro} - \mu_1(\sigma_{lro} + \sigma_{tro})] \\ \varepsilon_{tro} = \frac{1}{E_1}[\sigma_{tro} - \mu_1(\sigma_{\theta ro} + \sigma_{lro})] \end{cases} \tag{13}$$

The hoop strain of the outer wall of the patch tube is

$$\varepsilon_{\theta ro} = \frac{1}{E_1}\left[\sigma_{seql} - (1 - \mu_1)P_c\right] \tag{14}$$

Stress on the screen base pipe depends on the compression of the thick-wall cylinder and contact pressure $P_c$:

$$\begin{cases} \sigma_{tRi} = -P_c \\ \sigma_{\theta Ri} = \dfrac{R_o^2 + R_i^2}{R_o^2 - R_i^2} P_c = \dfrac{K_0^2 + 1}{K_0^2 - 1} P_c \end{cases} \tag{15}$$

Under the action of contact pressure $P_c$, the hoop strain for the inner wall of the screen base pipe $\varepsilon_{\theta Ri}$ is

$$\varepsilon_{\theta Ri} = \frac{1}{E_2} \left( \frac{K_0^2 + 1}{K_0^2 - 1} + \mu_2 \right) P_c \tag{16}$$

After patching, internal forming pressure $P_i$ is relieved. Residual contact stress $P_c^*$ ensures that the patch tube and the screen base pipe fit closely together in an elastic combination state. Ignoring the thinning of the patch tube, the stress on the outer wall of the patch tube is

$$\begin{cases} \sigma_{tro}{}^* = -P_c^* \\ \sigma_{\theta ro}{}^* = -\dfrac{D_i}{2t} P_c^* \end{cases} \tag{17}$$

Under the action of residual contact pressure $P_c^*$, the hoop strain for the outer wall of patch tube is

$$\varepsilon_{\theta ro}{}^* = -\frac{1}{E_1} \left( \frac{D_i}{2t} - \mu_1 \right) P_c^* \tag{18}$$

The stress on the inner wall of the screen base pipe is

$$\begin{cases} \sigma_{tRi}{}^* = -P_c^* \\ \sigma_{\theta Ri}{}^* = \dfrac{R_o^2 + R_i^2}{R_o^2 - R_i^2} P_c^* = \dfrac{K_0^2 + 1}{K_0^2 - 1} P_c^* \end{cases} \tag{19}$$

Under the action of residual contact pressure $P_c^*$, the hoop strain for the inner wall of screen base pipe $\varepsilon_{\theta Ri}{}^*$ is

$$\varepsilon_{\theta Ri}{}^* = \frac{1}{E_2} \left( \frac{K_0^2 + 1}{K_0^2 - 1} + \mu_2 \right) P_c^* \tag{20}$$

On the basis of the condition of deformation coordination

$$\varepsilon_{\theta ro} - \varepsilon_{\theta ro}^* = \varepsilon_{\theta Ri} - \varepsilon_{\theta Ri}^*, \tag{21}$$

substituting Equations (7), (14), (16), (18) and (20) into Formula (21) gives a relational expression for internal pressure, and residual contact pressure is:

$$\left[ \frac{1}{E_1} \left( \frac{D_i}{2t} - \mu_1 \right) + \frac{1}{E_2} \left( \frac{K_0^2 + 1}{K_0^2 - 1} + \mu_2 \right) \right] P_c^* = \left[ \frac{1}{E_1} (1 - \mu_1) + \frac{1}{E_2} \left( \frac{K_0^2 + 1}{K_0^2 - 1} + \mu_2 \right) \right] \left( P_i - \sigma_{seql} \ln k_0 \right) - \frac{1}{E_1} \sigma_{seql} \tag{22}$$

## 3. Hydroforming Properties of AISI 321 Patch Tube

### 3.1. Material Properties of AISI 321

To study the effect of forming speed on the strength and plasticity of the patch tube materials, tensile rod specimens composed of AISI 321 according to the shape and size specified in the standard (ISO 6892-1: 2009 Metallic materials—tensile testing—Part 1: testing method at room temperature). Unidirectional tensile tests were carried out on an electronic universal material testing machine (WDW-100 manufactured by Shandong Luda Testing Equipment Manufacture Co., Ltd., Tai'an, China). Tensile speed levels were set as 1, 2, 3, 4, or 5 mm/min. The comparison of the true stress–strain curves at different tensile speeds is shown in Figure 5. At a lower tensile speed, AISI 321 was stronger and had higher plasticity. Therefore, lower speed should be used when forming a patch tube.

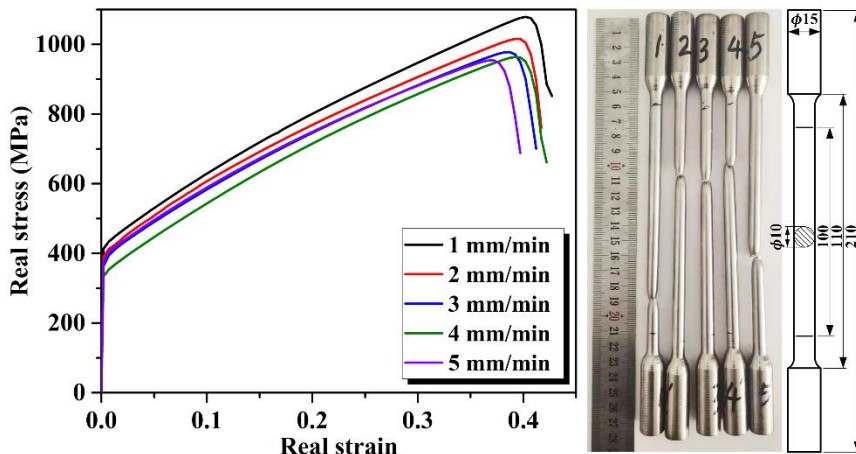

**Figure 5.** Stress–strain curves of tensile tests at different speeds.

As Figure 6a shows, the patch tube was prepared by some pretreatments: A stainless steel tube with 100 mm outer diameter and 6 mm wall thickness was first acid-pickled to remove the oxide layer on the surface, and evenly cold-drawn with lubricant many times until its outer diameter was 140 mm. Second, the solution treatment of the tube was carried out in an RC-120-11 continuous bright heat treatment furnace (made by Dongguan Fengda Industrial Electric Furnace Factory, Dongguan, China) [37]. Under the protection of the mixture of hydrogen and nitrogen, the tube was heated to 1080 °C for 60 min. The solution treatment can overcome the work hardening of the material, an uneven microstructure, and hidden machining defects caused by cold drawing. According to the shape and size specified in the standard (ISO 8496: 2013 Metallic materials—tube—ring tensile test) and the literature [38], arc and hoop tensile samples were cut axially from the patch tube (Figure 6b), installed on the tensile test machine, and subjected to tensile tests at a speed of 1 mm/min, as shown in Figure 6c–e.

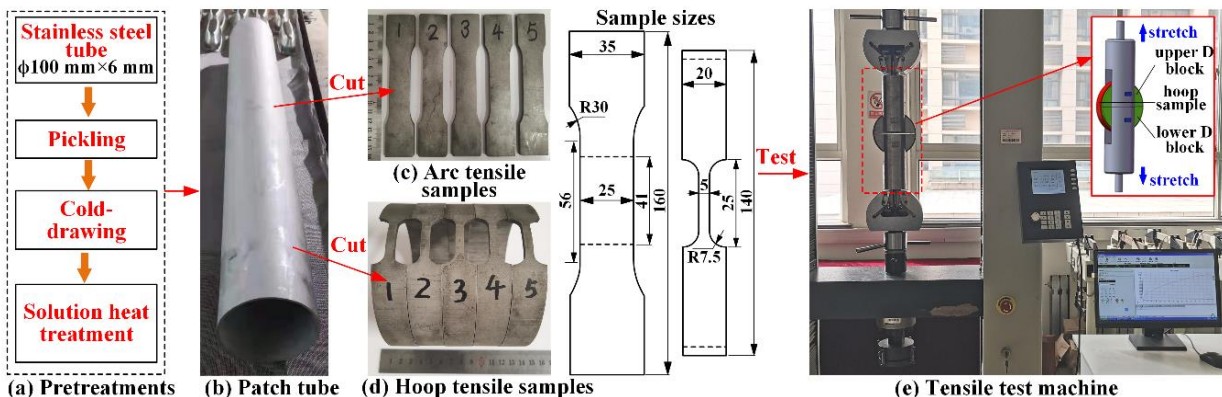

**Figure 6.** Preparation and mechanical properties analysis of AISI 321 patch tube.

The material mechanical properties (yield strength, tensile strength, and elastic modulus) were calculated from the uniaxial tensile data for multiple groups of arc samples. The strain strength coefficient and material hardening exponent were determined according to a fitting formula. The elongation of the pipe was calculated with multiple groups of data from the hoop tensile test. Some sample blocks (20 × 20 mm) were cut from the patch tube, leveled, the oxide layer was polished and removed, and samples were used for measuring the composition and hardness of AISI 321. As shown in Figure 7, composition analysis was conducted with the use of a full-spectrum spark direct reading spectrometer (GNR-mL300), and hardness was measured with a Micro Vickers Hardness Tester (THV-30HT manufactured by Lab Testing Technology (Shanghai) Co., Ltd. Shanghai, China). Tables 1 and 2 show the results of these calculations.

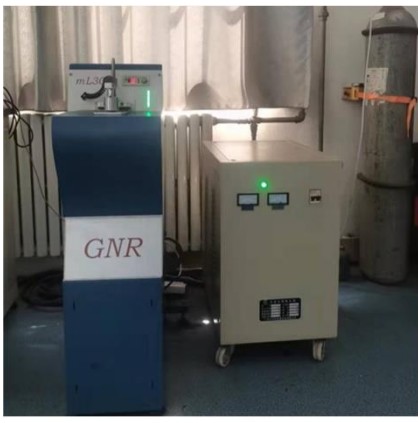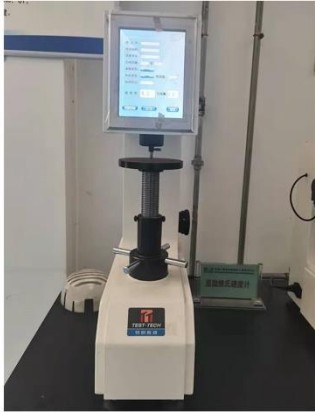

**Figure 7.** Composition and hardness analysis of AISI 321.

**Table 1.** Material mechanical property parameters.

| Material | Measuring Accuracy | Elastic Modulus $E$, GPa | Yield Strength $\sigma_s$, MPa | Tensile Strength $\sigma_b$, MPa | Poisson's Ratio $\mu$ |
|---|---|---|---|---|---|
| AISI 321 | ±0.5% | 190 | 278 | 702 | 0.290 |
| | | Strain strength coefficient $K$ | Material hardening exponent $n$ | Elongation $\eta$ | Hardness HBW |
| | | 1519.5 | 0.42 | ≥59% | 217.5 |

**Table 2.** Material composition.

| Element | C | Si | Mn | P | S | Ni | Cr | Ti | Fe |
|---|---|---|---|---|---|---|---|---|---|
| Content | 0.051 | 0.375 | 1.144 | 0.031 | 0.002 | 8.186 | 17.403 | 0.157 | 72.651 |

*3.2. Simulation of Hydroforming Patch*

On the basis of the patch hydroforming process and the strength requirements of the patch tube, the outer diameter of the patch tube should be 140 mm, and wall thickness should be 4 mm. A finite element model for hydroforming a patch tube with an outer diameter of 140 mm in a screen base pipe was established with Dynaform software, which was developed by corporations ETA and LSTC for the numerical simulation of sheet metal forming (as shown in Figure 8); a dynamic explicit algorithm was used to simulate the forming process. The model comprised a patch tube, screen base pipe, and upper and lower pushes (1 and 2). The patch tube was fixed at both ends, so it was divided into three parts: fixed, transition, and patch sections. The size of the patch section was: outer diameter × thickness = φ 140 × 4 mm and length = 1000 mm. The size of the screen base pipe was: outer diameter × thickness = φ 177.8 × 9.2 mm and length = 1100 mm. The size of pushes 1 and 2 is shown in Figure 8. The material model of the patch tube was based on the mechanical properties of AISI 321 (listed in Table 1).

For simplicity, the patch tube and screen pipe were modeled as rigid cylinders with no slots or central circular holes in simulations. The screen pipe and pushes (1 and 2) were also modeled as a rigid body and set as a fixed constraint [39,40]. Forming one-way surface to surface was chosen as the surface contact type between the patch tube and screen pipe. A three-parameter Barlat–Lian yield function was used for the constitutive model of the material (*MAT_36), which is suitable for the analysis of sheet metal plastic deformation and examining the thickness variation of thin-walled tubes after formation. In the meshing process, thin film units are used, and each unit is a quadrilateral surface with thickness. Due to large plastic deformation, the maximal mesh size of the patch tube was set to be 3 mm, and the minimal mesh size was set to be 1.0 mm in order to prevent the occurrence of grid penetration during simulation. For other parts of the model, the maximal mesh size was 20 mm, and the minimal mesh size was 5 mm. The mesh number was 54,142.

The solution time was set to be 0.6 s, and the time step was set to be $-1.2 \times 10^{-6}$. In the simulations of the forming process, some grid cells fixedly numbered at different locations were chosen as judgment nodes for observing the deformation, and patching was complete when the wall at the judgment nodes was no longer becoming thinner. Estimated total CPU time was 1 s, and estimated total clock time was 31,868 s.

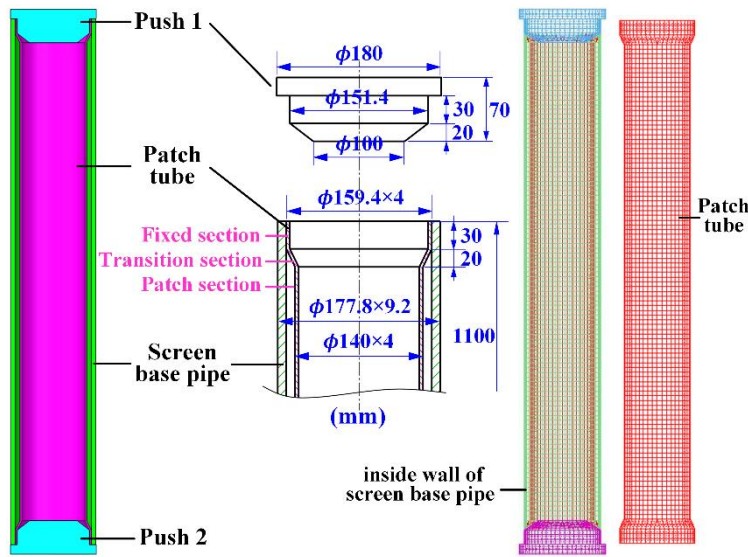

**Figure 8.** Model building and meshing.

To study the forming laws of the patch tube composed of AISI 321, the influences of the main structural parameters (length, initial outer diameter, thickness of patch tube) on hydroforming patch performance were simulated.

(1)     Length of patch tube

Because the length of a patch tube can affect pressure distribution in the tube and thereby forming performance, cases with different lengths (1000, 2000, 3000, 4000, 5000, and 6000 mm) were simulated, and results are shown in Figures 9 and 10. In the simulations, forming limit diagrams (FLDs) were obtained. Forming limit diagrams are the curves of the strip region formed by the real limit strain $\varepsilon_1$ (principal strain) and $\varepsilon_2$ (secondary strain) of the patch tube under different strain paths. On this basis, the forming property of patch tube could be accurately and effectively estimated. The area above the red curve is the crack zone where the sample is cracked and scrap. The area between the red curve and yellow curves is the critical zone, which indicates that the sample tended to crack, and it was easy for the failed forming parts to appear in this zone. The area between the yellow and green curves is the safe zone, which indicates that the sample could be smoothly formed to meet the technological requirements. Areas below the green curve are wrinkle zones (including wrinkle tendency, wrinkles, and severe wrinkles). This wrinkle defect is an obstacle to successfully repairing damaged screen pipes with tube hydroforming. According to the forming limit diagrams in Figure 9, all patch tubes could be fully formed, and there were no defects in the patch section after forming, but there were some cracks and wrinkle defects at both ends of the tube. With the increase in length, the area of cracks and wrinkles continuously expanded because both ends were set to be fixed sections in the simulations, and the expansion forming of the patch section can generate axial tension that acts on both ends and results in instability and thinning in the transitional section. Pressure distributions at 20%, 50% and 80% length of each tube are shown in Figure 10. Pressures at both ends (20% and 80%) were higher, and values were close, and pressure in the middle (50%) was lower. Pressure at each position did not obviously change with the increase in tube length. Results demonstrate that AISI 321 had good hydroforming properties.

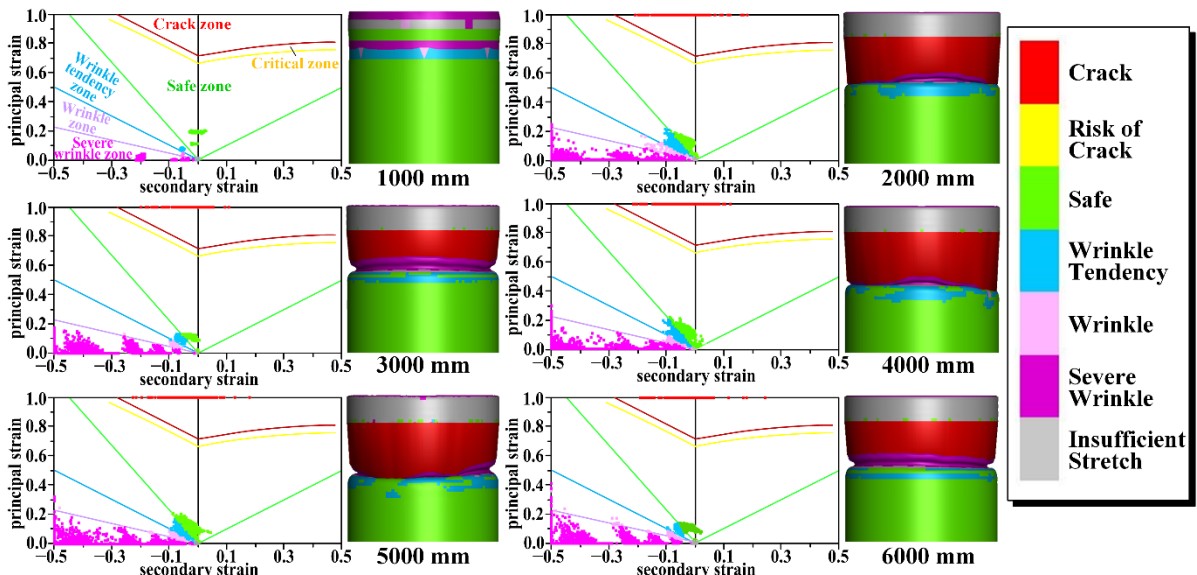

**Figure 9.** Forming limit diagrams of patch tube with different lengths.

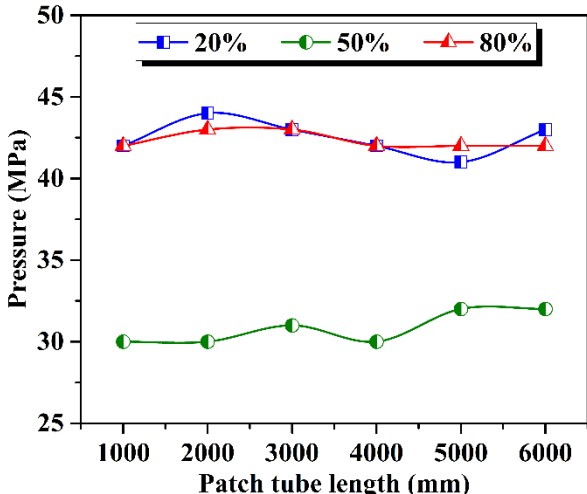

**Figure 10.** Pressure in patch tubes with different lengths.

(2)     Initial outer diameter of patch tube

The initial outer diameter of patch tube was set to be 128, 132, 136, 140, 144, and 148 mm. Simulation results are shown in Figure 11. With the increase in initial outer diameter, the forming pressure gradually decreased from to 46 to 38 MPa, and wrinkles and cracks at both ends of the patch tube tended to decrease. Thus, in order to reduce wrinkles and crack defects, and decrease forming pressure, a larger initial outer diameter of patch tube should be chosen.

(3)     Thickness of patch tube

In the simulations of the effects of the patch tube thickness on forming performance, thickness was set to be 2, 2.5, 3, 3.5, 4, 4.5, and 5 mm. Calculation results are shown in Figure 12. With the increase in patch tube thickness, forming pressure grew in an almost straight line, which shows that was increasingly difficult to realize the hydroforming patch. When the thickness varied from 2 to 5 mm, forming pressure increased from 14 to 58 MPa. At 5 mm, a severe wrinkle occurred in two transitional sections of the patch tube. Thus, in order to ensure the quality of the hydroforming patch and reduce forming pressure, patch tube thickness should be properly chosen.

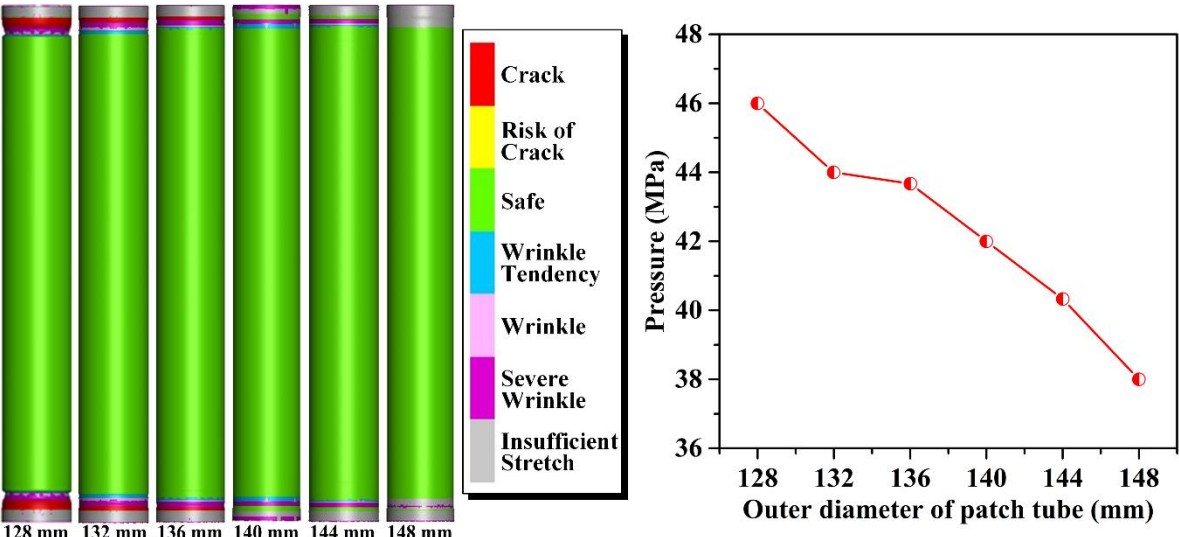

**Figure 11.** Influence of different outer diameters on forming pressure.

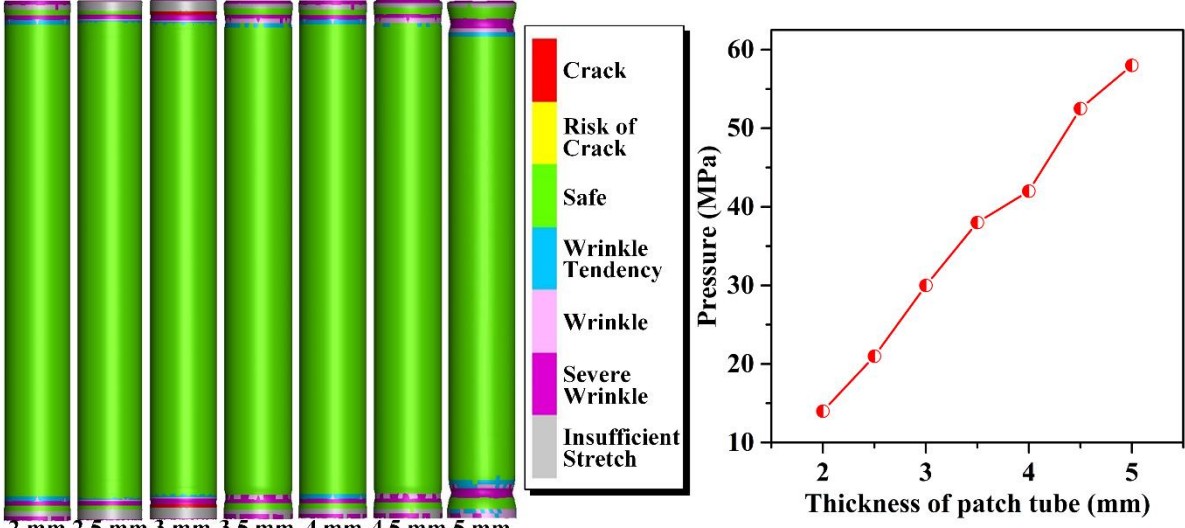

**Figure 12.** Influence of different thickness levels on forming pressure.

## 4. Test Verification

### 4.1. Hydroforming Test

In order to test the forming performance, a hydroforming test bench was built. A screen base pipe with 177.8 mm outer diameter, and an AISI 321 patch tube with 140 mm outer diameter and 4 mm wall thickness were used, as shown in Figure 13. During the experiment, water was injected into the patch tube with a hydraulic pump. Four strain sensors were arranged around the inner walls at each cross-section (at 20%, 50%, and 80% of the axial length) of the screen base pipe. Thus, a total of 12 strain sensors were used to detect forming pressure on the patch tube.

During forming, the 50% section was fully formed at 29 MPa, the 20% section at 38 MPa, and the 80% section at 40 MPa. Test results were slightly lower than the simulation results because pressure distribution in the patch tube was a bit uneven in the experiment. After the test, the screen base tube was cut every 200 mm, and the cross-sections were imaged, as shown in Figure 14. To avoid the influence of the push heads at both ends of the patch tube on the fit, the fit was only evaluated for the middle three sections. For each section, the screen base pipe and patch tube fit closely together.

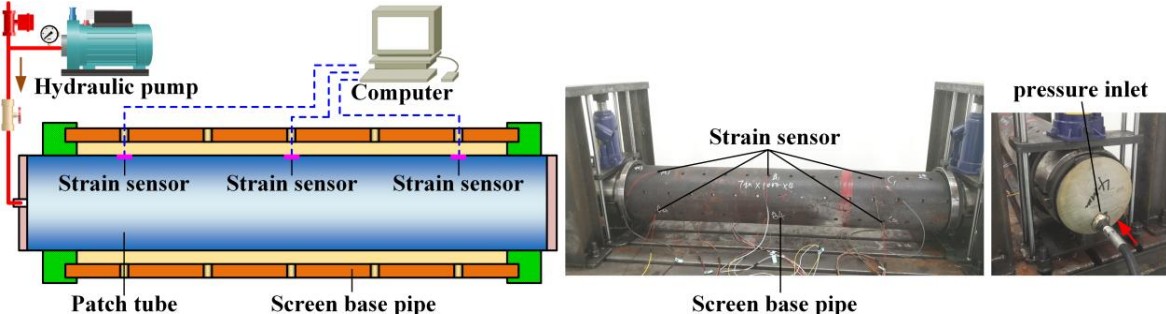

**Figure 13.** Hydroforming test of patch tube.

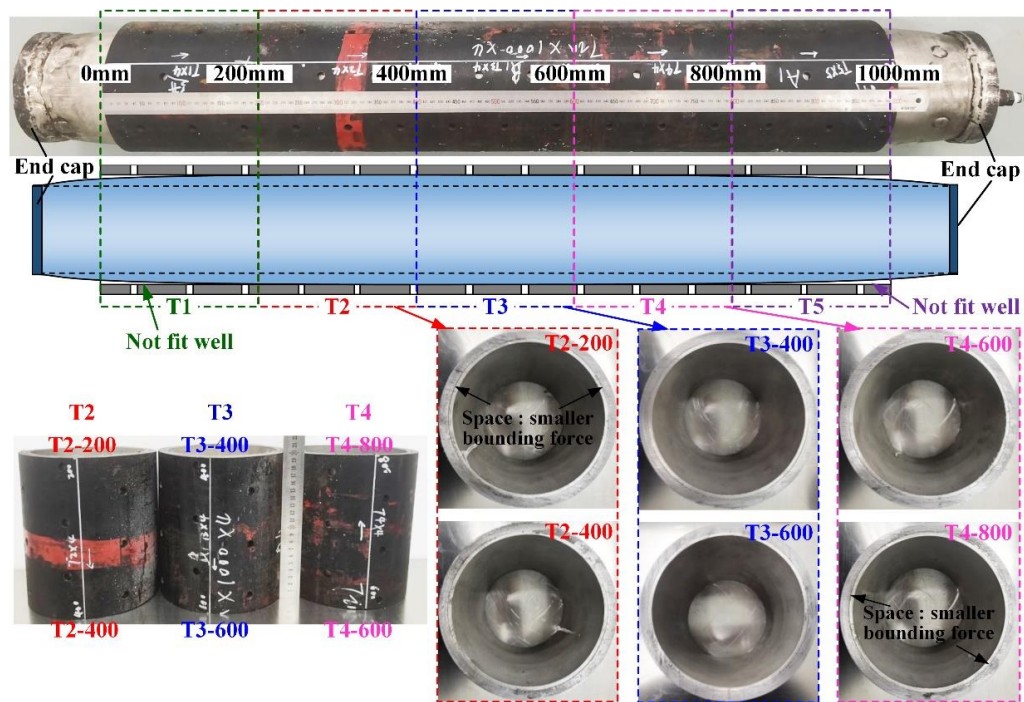

**Figure 14.** Hydroforming test results.

*4.2. Suspension Force Test*

When the patching was completed, there was residual contact stress between the patch tube and screen base pipe [41]. Suspension force is a measure of the quality of the formed patch that depends on friction between the patch tube and screen base pipe under residual contact stress. Greater residual contact stress can withstand greater suspension force, which means that the patch is fitted more strongly. The suspension force of the three sections of the patch tube was measured using the test device, as shown in Figure 15.

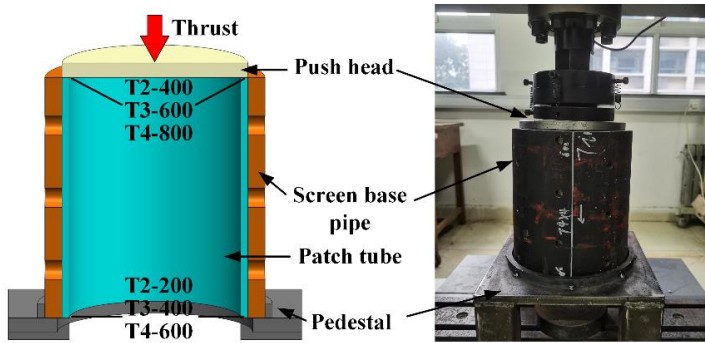

**Figure 15.** Suspension force test.

The measured suspension force is shown in Figure 16. The maximal suspension force for the T2 and T3 sections was about 25 kN, and about 14 kN for the T4 section, which met the requirements for a patch. The curves were mainly divided into static-to-dynamic conversion, growth, and steady stages. Within the displacement range of 0–25 mm, the T2 and T3 sections were still at the growth stage, and section T4 was at the steady stage. Suspension forces for the T2 and T3 sections were greater than that for T4, which indicates that the residual contact stresses for sections T2 and T3 were greater than that of T4, and that they were better-formed. During the experiment, the two ends of the patch tube were welded with the end caps, which caused the deformation of the two ends of the patch tube to be very small in the hydroforming process. However, the middle part of the patch tube fit closely with the screen base pipe, and its deformation was the largest. Thus, a certain conical degree in the outer diameter of patch tube was formed (that is, sections T1 and T5 of the patch tube did not fit well with both ends of the screen base pipe). Bonding strength at both ends was less than that in the middle; in section T4, the bounding force at the T4-800 side was smaller than that at the T4-600 side. In the experimental process, it was easier to separate the patch tube at the T4-800 side from the screen pipe, and its deformation is larger under the same pressure, while the displacement of T4-600 side remains unchanged. When the middle part of patch tube is prone to expansion and accumulation, which increases the friction force and causes a short and rapid rise in suspension force. After the separation of whole patch tube from the screen base pipe, the suspension force no longer rises, so overall suspension force was less than those for T2 and T3.

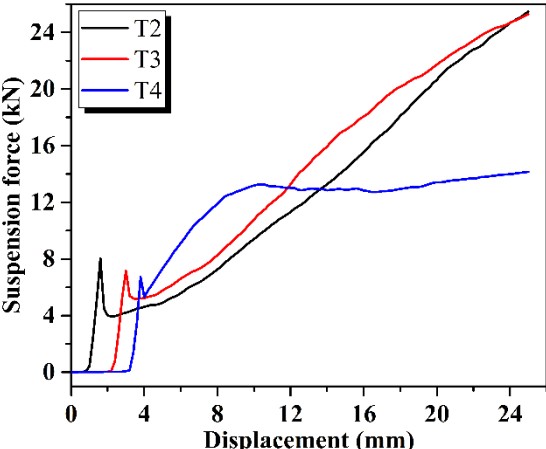

**Figure 16.** Suspension force curves.

Residual contact stress can be estimated by dividing the suspension load with the contact area between screen and patch tubes. Thus, residual contact stress after hydroforming was more than 139.78 $kN/m^2$.

### 4.3. Microstructural Analysis

Some samples (10 × 10 mm) were cut from the initial patch tube, the patch tube after solution treatment, and the patch tube after hydroforming. First, each sample was leveled and installed on a resin base. Then, samples were coarsely ground, finely ground, polished, cleaned, and dried for corrosion treatment. The blending ratio of the corrosive solution was: 40 mL hydrochloric acid (40 wt. %), 20 mL nitric acid (68 wt. %), 40 mL glycerol (15 wt. %), and 20 mL hydrogen peroxide (30 wt. %). Samples were etched for 20 min with the corrosive solution, neutralized with a saturated solution of sodium bicarbonate, and rinsed with anhydrous alcohol for microstructure analysis.

The patch tube was cold-drawn from a small-diameter AISI 321 pipe composed of Cr–Ni austenitic stainless steel. The microstructure showed that some of the austenite changed into deformed martensite during the cold drawing. Moreover, Ti refined the austenite grains and induced acicular ferrite. According to the microstructure of a lon-

gitudinal section of the patch tube, as shown in Figure 17a, chemical composition after cold drawing was uneven. The large deformation under the cold-drawing force limited the stable formation of austenite, resulting in serious fractures, so that the grains were small. Moreover, there were some banded structures along the deformation direction, the uniformity of the microstructure was poor, and the grain boundaries were disordered. The black spots or bands in the microstructure were pits formed by the loss of ferrite and carbide due to etching. Impurities were mainly $TiO_2$ and other inclusions [42].

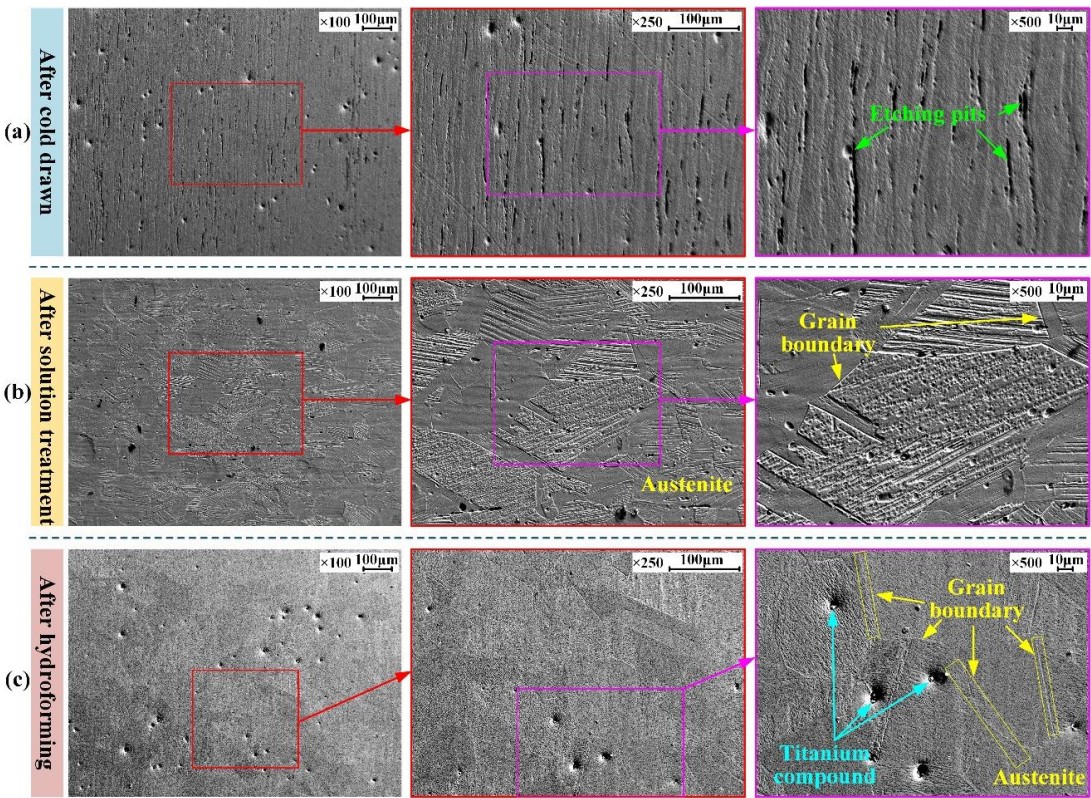

**Figure 17.** SEM images of patch tubes after different processes.

Figure 17b shows that, after solution treatment, carbides dissolved in the austenite, the microstructure was more uniform, the grain boundaries were clear, and the grains were coarse. The microstructure after solution treatment was mainly due to recrystallized austenite, and trace granular carbides and impurities.

As shown in Figure 17c, the microstructure changed little after hydroforming and was still dominated by austenite. The grain boundaries were still evenly distributed and clear. However, Figure 18 shows that the austenite morphology had a tensile trend along the deformation direction, and the deformed martensite preferentially nucleated at grain boundaries.

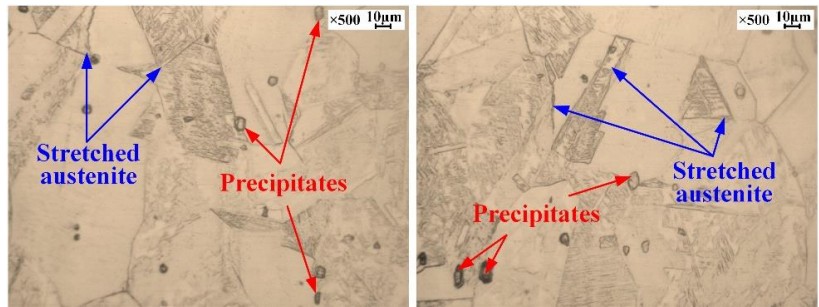

**Figure 18.** Optical microscope images of the patch tube after hydroforming.

Analysis of the microstructure of the cold-drawn patch tube shows that it had an uneven grain structure and underwent significant work hardening. There were deformed martensite and ferrite, which increased its brittleness and reduced its plasticity. Thus, high hydroforming pressure was needed, and defects could easily occur.

After solution treatment, the microstructure of the patch tube was mainly due to large austenite. The corrosion resistance and hydroforming quality are greatly improved, and the yield strength and tensile strength are lower, so a lower construction pressure could be used.

The plastic deformation of the patch tube due to hydroforming led to stretched austenite and the presence of some precipitates. Many studies [43,44] showed that, during the plastic deformation of austenite, there are some stress concentration fields at the grain boundaries. These fields are the first places of martensite nucleation. Further deformation transforms the retained austenite into deformed martensite, which can lead to the deformation strengthening and toughening of the material, and improve the strength of the patch tube after patching.

## 5. Conclusions

A technique for repairing damaged screen pipes based on tube hydroforming was presented, and its technical feasibility was investigated. Process and residual contact stress were analyzed. The mechanical properties, formability, residual contact stress, and microstructure of AISI 321 patch tube were experimentally investigated in this study. Results support the application of patch hydroforming. On the basis of the results, the conclusions can be summarized as follows:

(1) Mechanical property tests showed that AISI 321, after cold drawn and solution treatments, had better elasticity and plasticity at low forming speed.

(2) A numerical simulation model of a hydroforming patch composed of AISI 321 steel was built using of Dynaform software, and the effect of structural parameters, such as the length, initial outer diameter, and thickness of the patch tube, on hydroforming patch performance was investigated. Simulation results show that the forming pressure did not significantly change with an increase in patch tube length, but decreased with the initial outer diameter, and increased with thickness.

(3) A hydroforming test bench was constructed to experimentally test the patch method. Test results show that the patch tube could fit closely with the screen base pipe, and residual contact stress could be more than 139.78 kN/m$^2$, which meets the repair requirements for an underground damaged screen tube.

(4) The microstructures of patch tubes were compared and analyzed after different processes. Forming defects could easily occur in the cold-drawn patch tube. The solution treatment could effectively improve hydroforming quality and reduce construction pressure. Deformation strengthening due to hydroforming was conducive to improving the strength of the patch tube.

In order to achieve a better repair effect of a damaged screen base pipe, it is necessary to further screen or study patch-tube materials. Moreover, further research with respect to the hydroforming patch technology should be developed with the aim of field application.

**Author Contributions:** Data curation, J.C. and S.M.; investigation, Y.L.; methodology, H.W.; software, S.L.; supervision, H.W. and W.L.; validation, W.L.; visualization, Y.L.; writing—original draft, S.L.; writing—review and editing, S.L. and W.L. All authors have read and agreed to the published version of the manuscript.

**Funding:** This research was funded by the Fundamental Research Funds for the Central Universities, grant number 20CX02307A, and the Major Scientific and Technological Projects of CNPC, grant number ZD2019-184-004.

**Institutional Review Board Statement:** Not applicable.

**Informed Consent Statement:** Not applicable.

**Data Availability Statement:** Data are contained within the article.

**Conflicts of Interest:** The authors declare no conflict of interest.

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
