# Peer review of "Repairing Damaged Screen Pipes with Tube Hydroforming: Experiments and Feasibility Analysis"

_machines, doi:10.3390/machines10050391_

Round 1

Reviewer 1 Report

The paper presents an experimental-numerical study on using the hydroforming patch technology for repairing damaged screen pipes.

In general terms, the idea of the paper is solid, and its contents fit well within the scope of the journal. However, some issues should be addressed by the authors before the paper can be ready for publication. My suggestions and comments are as follows,

  1. In the last paragraph of section 1, the authors mention a previous study based on using AISI 321 patch tubes. Please indicate which work was this by including its reference.
  2. In section 3.1, please add here more information about the patch and screen tubes used in the experiments, namely dimensions, manufacturing processes and post-heating treatments if applicable.
  3. Were the uniaxial tensile tests on rod specimens (Figure 4) extracted from the tubes? If not, the results obtained here should not be applicable for the patch tubes because the mechanical behaviour will not be the same. Moreover, the elastic region is not correct because it shows an elastic modulus well beneath those of stainless steels (≈ 200 GPa).
  4. Please provide the tensile tests standards (or other sources) used for dimensioning the specimens shown in Figure 5. Ring hoop tests are not standardized yet, so a reference explaining the guidelines for carrying out these tests would certainly help strengthen the paper. For example: 10.1177/03093247211045236
  5. Did the tubes behave isotropically during the tensile tests? In order to clear out any doubts from future readers of your work, the flow curves of the ring hoop and longitudinal tensile tests could be presented to demonstrate its isotropic behaviour.
  6. Please provide equipment information and procedures used for obtaining the composition and hardness of the AISI 321. These would further increase the reproducibility of your work.
  7. Section 3.2 needs more information on the finite element models. Was the flow formulation used? What was the element number, CPU time and convergence criterion in a typical simulation? Please clarify also what are the “judgment nodes”.
  8. Were the simulations carried out with hexagonal elements? Why not use 2D models since hydroforming patching behaves pretty much in axisymmetric conditions having plane stress imposed in the longitudinal direction?
  9. Please improve the readibility of the images disclosed in Figure 7.
  10. Please discuss the much lower suspension force found in section T4 (Figure 14). Could it be related to the uneven pressure distributions verified in the hydroforming tests?
  11. The microstructure analysis work should present the methods used for preparing and etching the samples. The small dots shown in Figure 16 look a lot like Ti inclusions instead of austenite. Moreover, the microscopic images of Figure 16 should have a scale.
  12. In Section 5, the authors mention: ‘the residual contact stress could be more than 70 kN/m’ however this is not a stress unit. I would suggest estimating this contact stress by diving the suspensions load with the contact area between of the screen and patch tubes.
  13. I would also ask the authors to carefully proofread the manuscript. Exemples of misspelled terms: 
  • ‘7-inch patch tube’ – please use SI units
  • ‘Lam é’ – Lamé
  • ‘the clearance between the two tubes is closes’ – is closed
  • Subsection ‘4.2 Microstructure analysis’ should be 4.3

Reviewer 2 Report

Dear Authors,
Thanks for submitting the manuscript, see my comments below.

line 84: refer to the results of the research (previous study).

Fig. 1 (page 2) is only referred to on page 4 (after figures 2 and 3)

line 201, 214: the standard should be clarified

line 202, 223, 237, 248: give machine models and manufacturer's data as well as software model and manufacturer's data

Fig. 5 suggests a different title.

Table 1. Extend the obtained results with the error value

line 340: what did the metallographic preparation look like? how were the samples digested?

section 4.2. Microscopic analysis is questionable. TiN and TiC are not impurities. Does photo 15a show the microstructure of the metallographic specimen on the cross-section or surface of the pipe? If it is the surface of the plane, each corrosion pits results from the preparation of the metallographic plane and not from the hydroforming process. Similarly, "black spots or bands" formed as a result of alleged erosion. The term "fractures of the microstructure" is not authorized.

Regarding the organization of the text, more information and discussion of results is required in the Abstract. A negligible number of publications from MDPI was used in the introduction section and in the discussion of the results.

Reviewer 3 Report

The paper is devoted to a special tube hydroforming process designed to repair damaged pipes. The weakest point of the manuscript is its theoretical part. It is impossible to understand the assumptions and derivations.

  1. It is assumed that the expanding cylinder reaches a fully plastic state. It is necessary to justify that the assumption of small strains is acceptable
  2. Using Tresca’s yield criterion, one must justify the regime of flow chosen
  3. The mathematical definition of the equivalent yield strength is not provided. It is also necessary to show how this quantity has been calculated in the manuscript.
  4. The interface pressure is shown in Eq. (4) but does not appear in subsequent equations.
  5. Line 133. It is unclear why this solution is valid in the plastic region. Also, the strains are not provided in this equation or later. However, none of the principal strains vanishes, and it is necessary to separate the elastic and plastic strains.
  6. Line 166. What is generalized Hooke’s law?
  7. Finally, many solutions for the problem under consideration are available in the literature, even for more sophisticated constitutive equations. Some of these solutions can probably be used to replace the theoretical part of the paper. Or it is necessary to explain why available solutions are not appropriate. The links to some solutions are provided below.

https://doi.org/10.3390/pr9122161

https://doi.org/10.3390/met11050793

https://doi.org/10.3390/ma13132940

Round 2

Reviewer 1 Report

Dear authors,

All my comments and questions were addressed. Thank you for carefully revising your work.

Best regards

Author Response

We once again thank you for your valuable comments on our work.

Reviewer 2 Report

Dear Authors,
Thank You for submitting Your revised manuscript.

If the observed microstructures are the result of grinding, polishing and etching, then the effects observed in the microstructure will not be the effect of erosion (these effects should be removed during grinding), at most the effect of etching.

line 444: "and a small amount of ferrite," - please mark in the Fig.
What in figure 18 is indicated as "stretched martensite" corresponds in shape and size to what in figure 17 is described as "titanium compounds".

Reviewer 3 Report

Point 1 of my previous comment. There is no logical connection between this point and your answer. You use small strain solutions, such as the Lame solutions. However, you have not justified that the small strain assumption is reasonable. It has nothing to do with the plane stress assumption.

Point 5 of my previous comment. You have not commented on the separation of elastic and plastic strains.  

Round 3

Reviewer 2 Report

Dear Authors,
Thank You for submitting Your revised manuscript.

I still maintain that in Figure 18 the places marked as martensite nucleation are not martensite nucleations (please study the mechanism of martensite transformation), but rather related to titanium compounds or etching pits.
